fRNAkenseq: a fully powered-by-CyVerse cloud integrated RNA-sequencing analysis tool

Hubbard Allen 1 ahubbard@danforthcenter.org
Bomhoff Matthew 2
Schmidt Carl J. 3
1 Donald Danforth Plant Science Center , Saint Louis, MO , USA
2 Department of Plant and Soil Sciences, University of Arizona , Tucson, AZ , USA
3 Department of Animal and Food Sciences, University of Delaware , Newark, DE , USA
Gillespie Joseph
Electronic publication date: 2020 May 14
Publication date: 2020
Volume: 8
Electronic Location ID: e8592
Received 2019 Apr 11; Accepted 2020 Jan 18
Copyright: © 2020 Hubbard et al.
Copyright year: 2020
Copyright holder: Hubbard et al.
License: This is an open access article distributed under the terms of the Creative Commons Attribution License, which permits unrestricted use, distribution, reproduction and adaptation in any medium and for any purpose provided that it is properly attributed. For attribution, the original author(s), title, publication source (PeerJ) and either DOI or URL of the article must be cited.
License URL: https://creativecommons.org/licenses/by/4.0/

Keywords: CyVerse, RNA-Seq, Cloud integration

Funding: NSF Awards 1147029 and 1456942 USDA-NIFA-AFRI 2011-67003-30228 USDA National Institute of Food and Agriculture 2010-04233 This work was supported by the NSF awards 1147029 and 1456942 and USDA-NIFA-AFRI: 2011-67003-30228. This material is based upon work supported by the Agriculture and Food Research Initiative Competitive Grant No. 2010-04233 from the USDA National Institute of Food and Agriculture which also funded Allen Hubbard’s graduate salary. The funders had no role in study design, data collection and analysis, decision to publish, or preparation of the manuscript.

==============================
Background

Decreasing costs make RNA sequencing technologies increasingly affordable for biologists. However, many researchers who can now afford sequencing lack access to resources necessary for downstream analysis. This means that even as algorithms to process RNA-Seq data improve, many biologists still struggle to manage the sheer volume of data produced by next generation sequencing (NGS) technologies. Scalable bioinformatics tools that exploit multiple platforms are needed to democratize bioinformatics resources in the sequencing era. This is essential for equipping many research groups in the life sciences with the tools to process the increasingly unwieldy datasets they produce.

Methods

One strategy to address this challenge is to develop a modern generation of sequence analysis tools capable of seamless data sharing and communication. Such tools will provide interoperability through offerings of interlinked resources. Systems of interlinked, scalable resources, which often incorporate cloud data storage, are broadly referred to as cyberinfrastructure. Cyberinfrastructure integrated tools will help researchers to robustly analyze large scale datasets by efficiently sharing data burdens across a distributed architecture. Additionally, interoperability will allow emerging tools to cross-adapt features of existing tools. It is important that these tools are designed to be easy to use for biologists.

Results

We introduce fRNAkenseq, a powered-by-CyVerse RNA sequencing analysis tool that exhibits interoperability with other resources and meets the needs of biologists for comprehensive, easy to use RNA sequencing analysis. fRNAkenseq leverages a complex set of Application Programming Interfaces (APIs) associated with the NSF-funded cyberinfrastructure project, CyVerse, to execute FASTQ-to-differential expression RNA-Seq analyses. Integrating across bioinformatics platforms, fRNAkenseq also exploits cloud integration and cross-talk with another CyVerse associated tool, CoGe. fRNAkenseq offers novel features for the biologist such as more robust and comprehensive pipelines for enrichment than those currently available by default in a single tool, whether they are cloud-based or local installation. Importantly, cross-talk with CoGe allows fRNAkenseq users to execute RNA-Seq pipelines on an inventory of 47,000 archived genomes stored in CoGe or upload their own draft genome.

Introduction

RNA sequencing is a popular method to explore biological systems. It enables quantification of gene expression on a genome-wide scale. Along with other sequencing technologies, RNA-Seq has contributed to the “big-data” aspect of the genomics era. This means that it is also associated with challenges such as computational costs and data management burdens. Increasingly, these issues are bottlenecks in biological research. Although sequencing costs have decreased substantially for RNA-Seq, the computational footprint of RNA-Seq remains cumbersome. Much of this is due to the scalability hurdles of managing the large sample sizes and the number of experiments made possible by affordable sequencing (Papageorgiou et al., 2018). For example, RNA-Seq mapping and quantification alone requires gigabyte (GB) scale inputs in the form of reference files (FASTA and accompanying GFF annotations) as well as large libraries of read sequences which are often several GB, even when compressed. As output, read alignment produces burdensome alignment files which are also on the GB scale. Thus, even as many labs can afford next generation sequencing, maintaining the infrastructure and staff to process downstream data is expensive and access to core facilities varies across institutions. This problem will intensify in the near future as anywhere from 2 to 40 exabytes of data is anticipated to be needed to store human genome data alone by 2025 (Stephens et al., 2015), not including additional contributions by the growing plant and animal genomics communities. This data burden increasingly appears unmanageable without cloud driven resources (Navarro et al., 2019). Distributed computing systems, utilizing cyberinfrastructure, offer many advantages for bioinformatics workflows (Muir et al., 2016).

Cyberinfrastructure, loosely defined, describes a network of coordinated systems for computation and file management that empower the progress of data driven scientific research (Stewart et al., 2010). One strategy to comprehensively address the computational challenges of RNA-Seq and other NGS technologies is to create a generation of tools that easily manage large datasets by relying on cyberinfrastructure for each step of computation. These platforms would be able to share data and cross-talk through Application Programming Interfaces (APIs). APIs are protocols which allow the transmission of data across machines and platforms, often by standardizing data in the form of Javascript Object notation (JSON) form. An API-driven architecture would make it possible, through data sharing, for additional functionalities to emerge from the connection between tools, surpassing those present in any individual tool. The data burden placed on individual tools would be alleviated, as well. Such a forward-thinking architecture will enable tools to stay relevant as sequencing technologies evolve. This will allow developers to forge new workflows by joining features of different resources and create new platforms. This is the API driven approach taken by the tool fRNAkenseq. fRNAkenseq uses APIs associated with NSF-funded cyberinfrastructure to execute RNA-Seq analysis from read mapping to differential expression. fRNAkenseq provides biologists with two useful pipelines, one for read alignment and another for differential expression. Exploiting multiple forms of integration, fRNAkenseq leverages elements of NSF-funded cyberinfrastructure for both data storage and job management.

CyVerse: At the heart of fRNAkenseq is the cyberinfrastructure resource, CyVerse. CyVerse is a pioneering, NSF-funded project to make large-scale computing available to domain scientists (Merchant et al., 2016). Its offerings include a cloud-based data store and compatibility with publicly accessible APIs that allow developers to interact with CyVerse resources. Collectively, these offerings enable external groups to integrate and manage their own workflows on CyVerse platforms. Third party tools such as fRNAkenseq which are independent from CyVerse but which rely on CyVerse to a substantial degree are referred to as powered-by-CyVerse tools. CyVerse connectivity can be used to facilitate cross-talk between powered-by-CyVerse tools. For example, fRNAkenseq uses the central authentication system of CyVerse as a conduit to genome data in a separate powered-by-CyVerse tool, The Comparative Genomics Platform (CoGe) (Lyons et al., 2008).

Meanwhile, fRNAkenseq uses CyVerse compatible APIs to store and manipulate the large files associated with RNA-Seq analysis. This aspect of data movement across execution and storage systems (such as the CyVerse Data Store) is facilitated by the A Grid And Visualization Environment (Agave) API system (Dooley et al., 2012). The Agave APIs, which have spun off from the CyVerse development team, remain closely integrated with CyVerse and can coordinate data sharing between the CyVerse cloud data store and job execution machines. fRNAkenseq uses the Agave APIs to manage jobs and files across multiple machines and their implementation of the Agave APIs is described below. The Agave APIs are designed to enable science-as-a-service grid computing approaches. A grid computing strategy is one in which one or more execution and storage systems can be physically separated and still communicate to share data and tasks. This alleviates data burdens that would accumulate on a single machine. Grid computing is useful for scalable tool development.

The grid computing architecture in fRNAkenseq, made possible by Agave and CyVerse, enables data from multiple analyses to be stored in the central location of the CyVerse datastore. This is possible because a single CyVerse authentication token allows APIs to move data into and out of a user’s CyVerse account. As grid computing strategies become more popular for distributed architectures in bioinformatics, CyVerse will likely serve as a hub for an emerging generation of tools. APIs such as Agave serve as the glue which holds this network together. This enables the easy transfer of data across platforms. However, workflows designed by third party developers which make full use of this connected system are currently lacking. fRNAkenseq is one of the first tools to exploit the combined cyberinfrastructure capacities of CyVerse and Agave and the first to use CyVerses’s authentication system to cross-talk with CoGe. Other tools that have integrated with CyVerse infrastructure such as SciApps (Wang et al., 2018) are specific to DNA Seq alignment. Although the ecosystem of CyVerse resources has encouraged the development of powered by CyVerse tools such as iMicrobe (Youens-Clark et al., 2019), these offerings serve specific communities or integrate different elements of the CyVerse cyberinfrastructure environment than those utilized by fRNAkenseq. For example, tools such as iMicrobe do not include workflows that leverage both the Agave APIs and a separate set of APIs associated with an external third party tool, CoGe, to transfer and manage data. Integration with CoGe and the repurposing of this resource for RNA-Seq analysis in a third party tool, is an important aspect of fRNAkenseq’s novelty.

CoGe: CoGe is a tool originally developed for synteny analysis. CoGe uses an algorithm, called DAGchainer, to identify regions of conservation (synteny) in user selected genomes. To rapidly conduct synteny analyses, CoGe is built around a database of thousands of genomes. CoGe has existed since the early stages of the genomic era (2008) and has continued to evolve as computational approaches for NGS analysis have become more sophisticated. CoGe possesses its own suite of independent APIs which are nonetheless intertwined with CyVerse, as authentication into CoGe is accomplished through a user’s CyVerse ID. fRNAkenseq utilizes CoGe’s independently developed APIs which authenticate through a CyVerse user ID, in order to accesses the thousands of reference genome files in CoGe’s database and repurpose them for RNA-Seq analysis.

Informatics pipelines and features

fRNAkenseq is among the first of its kind to employ API-oriented integration with CyVerse from start to finish, but it also provides many important features of practical significance to biologists. Behind a polished graphical user interface (GUI) accessible to biologists, fRNAkenseq’s architecture threads together pipelines and resources that handle the various facets of RNA-Seq analysis from mapping (MapCount workflow) to differential expression analysis (DiffExpress workflow). The latter feature is significant because many experiments conclude with statistical analyses to detect differentially expressed genes across conditions. Importantly, fRNAkenseq’s DiffEpress workflow allows users to select genes that are differentially expressed by up to three unique differential expression tools, thereby enabling the selection of a robust candidate set of differentially expressed genes. Both the mapping and differential expression pipelines exploit a unique cloud-integrated architecture to ease the burden of RNA-Seq for biologists and fulfills a need for easy to use RNA-Seq pipelines that offer robust differential expression analysis.

It is important to note that the last few years have seen a diverse set of technologies developed to deal with the explosion of data in the sequencing era. Particularly regarding the many strategies to deal with RNA-Seq informatics, each resource comes with a set of advantages as well as disadvantages. Some desktop-based tools, such as RNA-Seq GUI (Russo & Claudia, 2014), cannot execute computationally intensive steps such as read mapping. Other server-based resources such as Galaxy offer a diverse array of tools, but are not oriented around sharing data across platforms. Alternatively, tools with an API-driven architecture that exploit CyVerse resources, such as SciApps (Wang et al., 2018), do not offer analyses specialized for RNA-Seq bioinformatics. SciApps does not facilitate data sharing with other powered-by-CyVerse tools such as CoGe. Importantly, fRNAkenseq fills an important niche, offering RNA-Seq pipelines that exploit CyVerse resources and share data across tools.

fRNAkenseq + Cyberinfrastruture: data science driven biology

fRNAkenseq and its API driven design leverages grid computing within CyVerse, setting it apart from non-server-based tools for differential expression such as RNA-Seq GUI. Meanwhile the connection with CoGe allows biologists access to thousands of genomes, with the ability to upload their own. Relying on the connectivity enabled by a proliferation of APIs within CyVerse and the powered-by-CyVerse tool CoGe, fRNAkenseq demonstrates how large-scale databases and infrastructure can be integrated to meet the RNA-Seq informatics needs of biologists and fuel a new generation of tools. In order to demonstrate the applicability of fRNAkenseq to real world data, we walk through an analysis which uses RNA-Seq to explore the heat stress response in poultry chicken. fRNAkenseq was developed in an animal genomics lab and although the tool is not species specific, tools like fRNAkenseq are particularly valuable to labs in the plant and animal sciences which often lack large-scale computational resources. The major cyberinfrastructure component of fRNAkenseq, CyVerse, resulted partly from an NSF driven consolidation of earlier iPlant and iAnimal initiatives, which were cyberinfrastructure services for the plant and animal communities, respectively.

Workflow and Architecture

Architecture: Agave API utilization for mapCount and diffExpress

The fundamental software unit in the Agave API system underlying fRNAkenseq’s backend is an Agave App (Fig. 1A). An App posted to the Agave API service consists of a shell script template with its JSON wrapper. Agave Apps can move data freely in and out of the CyVerse datastore. Agave Apps make it easy for a tool like fRNAkenseq to manage multiple complex workflows. This is because each workflow can be instantiated as a separate app in the Agave service. This modular approach allows the computationally intensive mapping pipelines of MapCount (Fig. 1B) to be encoded as an Agave App separately from the differential expression pipelines DiffExpress (Fig. 1C).

Figure 1 Data lifecycle of fRNAkenseq job and schema illustrating how components across CyVerse and CoGe relate to one another.

(A) Outline of Agave App, (B) schema for MapCount, (C) schema for DiffExpress, (D) outline for connectivity with fRNAkenseq and CoGe as well as Data Store. User data uploaded into CyVerse is accessible for analysis by the fRNAkenseq front end. The path followed by user uploaded sequencing data during an analysis is illustrated in the green arrows.

DiffExpress is also instantiated as an Agave App. Agave Apps are flexible enough that the multiple algorithm R-based differential expression pipeline of DiffExpress is easily coded in a single workflow. As part of fRNAkenseq, these pipelines are accessible through a polished user interface, which also allows users to select CoGe genomes for alignment and quantification (Fig. 1D). Other resources which utilize Agave, such as SciApps (Wang et al., 2018), do not offer differential expression pipelines or leverage cross talk with CoGe. The tool cross-talk with CoGe is useful because it expands the number of genomes a user can use to analyze RNA-Seq data.

Workflow

fRNAkenseq is divided into two major modules accessible through separate interfaces: One for read mapping and transcript quantification (MapCount) and a second for Differential Expression (DiffExpress). Both use the Agave APIs to move data across CyVerse cyberinfrastructure. The computationally intensive pipeline MapCount relies on two state-of-the-art algorithms, Hisat2 and Stringtie (Kim, Langmead & Salzber, 2015), using generally default settings. Typical workflows take users through both MapCount and DiffExpress. Users will quantify genes with Mapcount and subsequently complete differential expression using DiffExpress. The implementation of these pipelines is described below, demonstrating how both depend on connection with CyVerse and the Agave Apps API system.

MapCount

MapCount relies on the lightweight mapper HiSat2 with default settings for mapping. Samtools (Heng et al., 2009) is used to convert these files into a compressed .bam file using samtools view -bS, A .bam file is subsequently sorted using samtools sort. StringTie is subsequently used to determine FPKM and TPM, with non-default parameters: -G -eB. Per the StringTie manual, counts are determined using the prepDe.py script provided from http://ccb.jhu.edu/software/stringtie/index.shtml?t=manual. The HiSat2 and StringTie workflow was selected as the backbone for fRNAkenseq’s mapping and quantification (Pertea et al., 2015) pipeline because these tools, as successors to the popular Tuxedo pipeline (Trapnell et al., 2012) are well established and computationally efficient. During MapCount, genome reference files (FASTA and annotation) are automatically pulled from CoGe using an API call. These reference files are indexed prior to alignment, if an index does not already exist. Indexed reference files are cached in a user’s CyVerse directory to avoid redundant computation time.

DiffExpress

Frequently, one of the next steps for RNA-Seq analyses involving samples is the determination of genes differentially expressed between the control and treatment groups. DiffExpress was developed to make this step simple, rapid and robust. DiffExpress enables rapid and stringent differential epression analysis. It deploys multiple algorithms: edgeR (Robinson, McCarthy & Smyth, 2010), DESeq2 (Love, Huber & Anders, 2014), BaySeq (Hardcastle & Kelly, 2013) in a single pipeline. DiffExpress combines the outputs of these algorithms into gene lists of varying selectivity according to the number of tools that have determined a gene to be differentially expressed. Standard workflows in DESeq2 and edgeR are run, corresponding to the general steps of normalization and statistical testing. Differentially expressed genes are identified at the p < 0.05 level.

DESeq2

The first DESeq2 function used in the DiffExpress pipeline is DESeqDataSetFromMatrix(), using default parameters with the design parameter equal to a vector denoting sample membership in control or experimental conditions. Subsequent differential expression is accomplished with the functions DESeq() and results(), also using default parameters.

edgeR

edgeR first uses the DGEList() function to create an edgeR object. Normalization is accomplished using calcNormFactors() and estimateCommonDisp() with default parameters and differential expression determined using exactTest().

BaySeq

BaySeq uses the function getLibsizes() for normalization getPriors.NB() to calculate prior probabilities for differential expression and parameter settings, samplesize = 1,000, estimation = “QL”. Likelihood of differential expression is calculated using getLikelihoods.NB() function with parameter settings pET = “BIC”.

This set of multiple algorithms for differential expression was selected because of the popularity and diversity of statistical strategies represented in the default approaches of these tools. For example, while all of these algorithms model read distribution using a negative binomial model, each has specialized approaches to calculating the relevant parameters. Computational studies support using multiple tools to create a more robust differential expression pipeline, as differences between individual tools generate distinct sets of differentially expressed genes. For example, previous work which analyzed the output of multiple RNA-Seq packages across a complex dataset used hierarchical clustering to show that edgeR and DESeq2 perform similarly, while BaySeq demonstrates distinct performance profile compared to these two tools (Schurch et al., 2016). The performance of each tool can vary substantially based off of such factors as sample size and sample-to-sample variability (Seyednasrollah, Lahlo & Elo, 2015). For example, changing default parameters of edgeR may improve performance in some datasets (Hardcastle & Kelly, 2010). while decreasing performance in others. Additionally, it is known that DESeq2 predictions may be skewed when a large fraction of genes are influenced in the same direction (Hardcastle & Kelly, 2010).

Ultimately, genes which are found to be differentially expressed according to only a single tool, but not others, have been found to be much more likely to be false discoveries (Tan et al., 2015). Thus, by analyzing the intersection of predictions from multiple enrichment programs with fRNAkenseq, it is possible to choose a more robust set of differentially expressed genes than would result from a single tool alone. In terms of practical guidance for biologists to prioritize genes, those which are differentially expressed according to all three tools are a top priority for follow up. These are identified by default in the DiffEpress workflow. For cases in which the set of genes predicted as differentially expressed by multiple tools is too small, we have found that as a single tool, BaySeq often performs the most stringently in small to medium sample sizes (2–10 replicates). This is likely because the empirical sampling technique of BaySeq (Hardcastle & Kelly, 2013) prevents the variability introduced by genes with extreme values from skewing parameter estimation.

After each tool (DESeq2, edgeR and BaySeq) is run in the DiffExpress pipeline, lists of differentially expressed genes output by each package, as well as those that overlap between two or more packages are created. This feature is important, because despite many approaches to differential expression, none is found to be universally superior (Rapaport et al., 2013) for all experimental designs. An implementation of fRNAkenseq, through MapCount and DiffExpress is provided below.

Demonstration

Here, we demonstrate a complete execution of fRNAkenseq, from genome loading through CoGe and FASTQ loading through the CyVerse datastore, all the way to differential expression. The dataset is from a previously published study (Jastrebski, Lamont & Schmidt, 2017) that explores the effect of chronic heat stress on commercial poultry chickens, from the perspective of the liver transcriptome. It is comprised of two samples from chickens raised under control conditions (GSM2496017 and GSM2496018) and two samples from chickens raised under heat stress (GSM2496032 and GSM2496031). This study used temperature-controlled bird houses at the University of Delaware to heat stress broiler chickens. Control birds were raised in one house and heat stressed birds were raised in a separate house. The control house was maintained at a thermoneutral 25 °C. Chickens in the heat stress house were heat stressed for 8 h per day for 1 week, during which time the temperature was increased to 35–37 °C and liver samples were collected at 28 days post hatch (Jastrebski, Lamont & Schmidt, 2017). This type of small but controlled study is typical of a pilot study for a user who would be a potential user for fRNAkenseq.

Genome loading into CoGe and read mapping

If a genome does not exist in the CoGe database, a user will upload it into CoGe for analysis with fRNAkenseq (Fig. 2). This is done by uploading both the FASTA and annotation associated with an organism. Genomes uploaded into CoGe are accessible by fRNAkenseq because of the cross-talk facilitated by the CoGe APIs. Users select FASTQ libraries uploaded into the CyVerse data store (Fig. 2) for read alignment and transcript quantification in MapCount (Fig. 3). As a reference genome for mapping, they may select any of the 40,000 + genomes loaded into CoGe, or the draft genomes which they have manually uploaded. The user will select the desired genome to run the MapCount pipeline. For this demonstration, the chicken genome used is Gal-Gal5, secured through NCBI. After mapping is complete, users will then be able to execute DiffExpress, to conduct differential expression pipelines and identify genes which differ between conditions. Users may also choose to evaluate alignment files using CoGe’s genome browser utilities.

Figure 2 Genome uploading into CoGe.

Genome uploaded into CoGe for use in fRNAkenseq using CoGe’s FASTA and annotation loading tools.

Figure 3 Uploading FASTQ through CyVerse data store and MapCount.

Users upload their FASTQ files using the CyVerse Data Store (https://de.cyverse.org/de/). fRNAkenseq accesses FASTQfiles in a user’s directory using a files-list query from the Agave API service. All genomes in CoGe are available to map loaded FASTQ files against using fRNAkenseq MapCount. Genomes can be identified by their unique ID number. Genome 52438 which was uploaded through CoGe is now accessible for RNA-Seq analysis with fRNAkenseq.

DiffExpress comparison

As previously described, each of the algorithms in DiffExpress (edgeR,DESeq2, BaySeq) have advantages and drawbacks regarding sensitivity and specificity. The landscape of tradeoffs will vary based on noise and sequencing depth on a gene to gene basis (Rapaport et al., 2013). The output of a sample analysis, show that there is an abundant, but reduced, set of genes at the center of the Venn Diagram (Fig. 4). These genes would be top priority for follow-up for further computational analyses or targeted methods, depending on the purpose of the study. If the number of differentially expressed genes at the center of the Venn Diagram is not sufficiently large such that a user wished to focus on genes selected by only a single tool, we suggest genes identified as differentially expressed by BaySeq may serve as high priority for follow up. Users may also determine the desired degree of stringency by prioritizing genes which are differentially expressed according to two or all three of the algorithms. When users intend to follow up on candidate genes with expensive, low throughput validation methods, it is recommended that candidate genes which are differentially expressed according to two or more tools are selected.

Figure 4 DiffExpress output: predicted differentially expressed genes by tool.

Number of genes predicted to be impacted by heat stress from fRNAkenseq DiffExpress according to the number of tools which predict differential expression. Eighty-one genes were differentially expressed according to all three programs.

Conclusions

fRNAkenseq utilizes CyVerse and another resource connected to CyVerse, CoGe, to remove data management and computational burdens associated with transcriptome analysis. Through cross-talk and data sharing, fRNAkenseq extends the usefulness of both CyVerse and CoGe, while offering novel features to biologists, such as a useful pipeline that executes differential expression with multiple tools. fRNAkenseq is built around the idea that in the future, bioinformatics tools will need to evolve with the increasingly complex workflows biologists use to analyze NGS data. Cross-talk and modular designs will enable data sharing so that complex analyses can be executed across bioinformatics platforms, thereby expanding the capacity of each individual, connected resource. For example, because fRNAkenseq uses the CyVerse Data Store as a data repository, files can be shuttled between fRNAkenseq and other powered-by-CyVerse tools or Apps in the CyVerse Discovery Environment. Although these analyses are beyond the scope of a typical RNA-Seq analysis such as that conducted in the demonstration, other tools in the CyVerse computing environment, such as those in the Discovery Environment, can also be used to analyze files produced by fRNAkenseq. Users who wish to execute more statistically advanced differential expression analyses than those implemented by DiffExpress, may use the CyVerse Discovery Environment Apps, DESeq2 and edgeR which offer additional options for complex experimental designs. Future development will focus on furthering cross-talk between fRNAkenseq and other resources such as those offered by the open science grid (OSG) (Pordes et al., 2007) and Jetstream (Stewart et al., 2015).

fRNAkenseq’s deployment of an API driven architecture from start to finish demonstrates the advantages of this design from several engineering standpoints. For example, by relying on the CyVerse cloud data store, fRNAkenseq avoids the creation of additional systems for file management. Manipulation of data in this environment is accomplished by the Agave API system, which ferries data to an external execution machine, before returning data back into the CyVerse data store. Thus, the data bottlenecks which arise from leveraging the same machines for data storage and execution are avoided by fRNAkenseq. Integration with CyVerse also makes it possible to leverage the thousands of genomes stored in CoGe for RNA-Seq analysis because CoGe’s data transfer APIs (which are independent from Agave but integrated with CyVerse) are nonetheless accessible with a single CyVerse authentication token.

Although existing platforms, such as Galaxy, offer pipelines for biologists to execute RNA-Seq analysis, they do not leverage cross-platform workflows such as those utilized by fRNAkenseq. Unlike Galaxy, fRNAkenseq accesses the powered-by-CyVerse tool CoGe and makes all genomes in CoGe’s database available for RNA-Seq analysis. Because of the cross-talk between CoGe and fRNAkenseq, a user may also upload a genome into CoGe for analysis in fRNAkenseq. This makes RNA-Seq possible when users possess only a draft genome. This is a common situation for those studying non-model organisms. The connection with CoGe also allows easy access to CoGe’s genome browser, enabling users to easily examine read alignment files. While fRNAkenseq is intended only for reference-based RNA-Seq alignment, users have the freedom to prepare their own de-novo references which can be uploaded into CoGe. These users may use other tools to conduct reference free alignments, but still use fRNAkenseq’s MapCount pipeline in order to compare the agreement of reference free approaches to reference-based one. This could be important in the context of emerging reference free RNA-Seq pipelines, as reference free approaches (while computationally lightweight) can have significant issues quantifying lowly expressed transcripts and small RNA’s (Wu et al., 2018). By leveraging state of the art cyberinfrastructure and emerging APIs, fRNAkenseq produces a tool useful to biologists and provides a blueprint for the next generation of cloud-based informatics resources which will be produced by CyVerse.

Additional Information and Declarations

Competing Interests

Author Contributions

Data Availability

The authors declare that they have no competing interests

Allen Hubbard conceived and designed the experiments, analyzed the data, prepared figures and/or tables, authored or reviewed drafts of the paper, and approved the final draft.

Matthew Bomhoff performed the experiments, analyzed the data, authored or reviewed drafts of the paper, and approved the final draft.

Carl J. Schmidt conceived and designed the experiments, prepared figures and/or tables, authored or reviewed drafts of the paper, and approved the final draft.

The following information was supplied regarding data availability:

Raw data is available at GitHub: https://github.com/victorfrnak/hello_world_frnak.

Data is available at NCBI GEO: GSM2496032, GSM2496031, GSM2496017 and GSM2496018.

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
