# Peer review of "fRNAkenseq: a fully powered-by-CyVerse cloud integrated RNA-sequencing analysis tool"

_PeerJ, doi:10.7717/peerj.8592_

## Round 0.1 · original submission · Major Revisions

Dear Dr. Hubbard and colleagues:

Thanks for submitting your manuscript to PeerJ. I have now received two independent reviews of your work, and as you will see, the reviewers raised some concerns about the research. Despite this, these reviewers are optimistic about your work and the potential impact it will have on research communities studying large-scale transcriptomics datasets. Thus, I encourage you to revise your manuscript accordingly, taking into account all of the concerns raised by both reviewers.

Aside from the many suggestions raised by the reviewers, there is general consensus that a reorganization of the manuscript is needed. Please follow the suggestions of the reviewers regarding this, and please ensure that you enlist the help of an outside reader to catch the many typos that occur throughout the manuscript.

Also, a complete and detailed overview and implementation of the pipeline, as well as the functionalities that make it unique relative to other tools/platforms, is lacking. This must be included in your revision.

I look forward to seeing your revision, and thanks again for submitting your work to PeerJ.

Good luck with your revision,

-joe

·

Basic reporting

The author are presenting the workflow fRNAkenseq that runs on the CyVerse platform. This workflow allows to manipulate RNA-Seq sequencing data towards the identification of differentially expressed genes using some standard protocols (but not a RNA-Sequencing platform as expected from the title).

The introduction needs to be improved by a state of the art of the tools that carry on differential expression from RNA-Seq data, and a description of the advantage of Cyverse in a user perspective manner (for instance its facility to upload and store data or to link different tools), and definitely why a workflow for managing those data in CyVerse is appealing.

In a second part of the article, a complete and detailed overview of the pipeline, how it can be used, the (new) functionalities it is offering, its input, parameters and output, is lacking.

I think that is there is a confusion between enrichment and differential expression analyses. Overall, the article is not properly written, lot of typo are remaining (even the name of the tool is often misspelled). The manuscript needs definitely to be corrected or rewrote correctly. There is too many figures, most of them are screenshots not well described (in the text and in the legend).

Experimental design

The authors focus on the comparison with Galaxy, which should not be the purpose of the article. Indeed, because the fRNAkenseq has been developed in the CyVerse environment, which is probably good because such pipeline was demanding in CyVerse, and also because that it is probably possible to derive fRNAkenseq in Galaxy, as well, by adjusting the tools parameters and adding few other functionalities or wrapped scripts such as the comparison of the results from the various differentially expressed tools.

But again, the purpose of this article is necessary to present more clearly how the pipeline is working, from a user point of view, to propel biologists to use it instead of creating their own pipeline in Galaxy. First of all, I think it is important to show why the use of 3 differential expressions tools would help to define a more specific list of genes in a valuable example. Then give a more detailed the description of why and how the links to CoGE is useful. On the other hand, I would expect that the advantage of such tools in the CyVerse platform should facilitate the analysis of enrichment in Gene Ontology terms. This still could probably be well illustrated by use of an example.

Validity of the findings

The principal arguments of the authors is that there is no similar pipeline in Galaxy, so far, and at least no one which "synthesize the output of multiple enrichment prediction programs into easy to interpret tab delimited files". But an experienced Galaxy user should probably be able to perform this using the tabular tools. Morevover, the sentence l99-100 "steps of the fRNAkenseq pipeline which cannot be implemented on the Galaxy platform and would otherwise need to be executed manually (for
example combining the output of multiple tools)." is not exact, because such tool or script could be rapidly developed and wrapped into Galaxy. On the other hand, the sentence l189-l200, "Galaxy commands by default for DESeq2 are likely more stringent for those used by fRNAkenseq" is not demonstrative because the default parameters can be changed in Galaxy, by the administrator, and parameters can be changed by user before running or when creating a workflow.

As a conclusion, comparing fRNAkenseq to Galaxy is not pertinent, because one of the most advantage of Galaxy is that it can be installed and configured in many places, where people can develop their own version of their favourite pipelines and distribute them if they want. By the way, there is also some way to create Cyverse workflows into Galaxy, and making fRNAkenseq working on both platforms should probably be feasible. Nevertheless, the advantage of CyVerse is that they are offering a platform on top of very large resources.

So, I would recommend to move the Galaxy and Cyverse comparison to the discussion, because it is not a central point.

Additional comments

It is important to ease workflows either in Cyverse, SciApps or Galaxy to be published elsewhere in order to increase their usage and to be cited. This article shows a interesting pipeline that allow users to perform a complete RNA-Seq analysis using a standard mapping strategy and 3 different popular tools for differential expression. The main part of the article consists in comparing fRNAkenseq to an almost similar pipeline generated through Galaxy, while the principal object of this article should be a demonstration of what the pipeline is achieving in a user perspective, enhanced with some technical aspects of the pipeline, and perspectives for a public of computer scientists.

So, in order to increase its readability and probably its impact, I would encourage the authors to reorganize the writing of this manuscript, by simplifying the comparison with Galaxy and moving this part to the discussion, while focusing in the main part to the usage of the pipeline in the CyVerse environment, with a better and clearer illustration, and some technical considerations.

·

Basic reporting

The manuscript presents an overview of a workflow for expression studies, RNA-Seq in particular, that is presented in the context of Cyverse and CoGE. The workflow seems to be a valuable contribution to the community however the key details are not presented in enough detail for a reader to evaluate this resource. The authors should provide more literature to describe the best practices in expression analysis for short read data and provide context to the tools selected. They should also review the limitations of this approach (reference genome only?). The organization of the paper is problematic in that, key technologies are not introduced early on and important details are scattered through the paper. The figures are too numerous and not helpful (i.e. screenshots of gene expression outputs). I have provided some feedback on these. Finally, the comparison with Galaxy is executed with a sample RNA-Seq study however, the results are presented in basic counts of genes and there is no benchmark for this so the emphasis here is not helpful in terms of evaluation. It would make sense to focus on features, hardware, runtime, flexibility, etc that one would associate with workflow comparisons.

Experimental design

The paper should focus on the details on the workflows (more detail) and less detail on the sample study and sets of gene counts. The limitations and advantageous of both frameworks should be presented. Tables comparing features, tools, downstream merging of results, etc would be more helpful to the reader.

Validity of the findings

A restructuring of the background and workflow details will allow a better evaluation of this framework. I believe it is quite useful but it needs to be explained in more detail and accessible to a broad readership not already familiar with CoGE, Galaxy, or even Cyverse.

Additional comments

Please note the following:

Abstract:
(1) The abstract could be much more informative about the specifics of the problem and the solutions proposed. Are datasets for RNA-Seq getting larger? I don’t think that is really the case - the read per library target has been established for some time, and is not particularly high for species with a reference genome. You are addressing specific challenges but these are not well described here. I should be very clear why this was developed after reading the abstract.

(2) This sentence needs to be restructured “A strategy to counteract this challenges is developing a modern”

(3) I would avoid citations in the abstract.

(4) The last sentence of the abstract, “Cross-talk with another CyVerse associated tool, CoGe (Lyons et al., 2008), meanwhile,
allows fRNAkenseq users to execute these pipelines on an inventory of 47,000 genomes or upload draft
genomes.” needs to be re-written.

Introduction
(5) The introduction is structured as just one paragraph but should be sub-divided into key sections that would outline the background the reader needs to understand this tool and its benefit. You go on to compare SciApps, Galaxy etc - these should be at least briefly introduced in the introduction within the scope of cyberinfrastructure. One potential organization might include:
(a) Cyverse and other cyberinfrastructure
(b) Introduce CoGE
(c) High level overview of challenges with RNA-Seq informatics
(d) Challenges addressed by this application

(6) I would recommend describing the pipeline in detail before moving into the Architecture section. The applications used in the pipeline would not really fall into the architecture section.

(7) In describing the pipeline applications, I would provide more detail on their selection, role, and possible parameterization. Literature describing the current best practices for RNA-Seq analysis should be referenced here.

(8) Line 89-90 - “This feature is important, because despite many approaches to differential expression, none is found to be universally superior”. This is not completely accurate - many of the R packages are recommended in different study designs. It would be advantageous for the workflow to provide a recommendation for the appropriate applications based upon the replication model.

(9) Figure 1: Data Store is a Cyverse product and should be represented in caps and consistently in Figures, legends, and in text.

(10) Figure 1 should be refactored to clearly indicate the order of operations and where data is initially loaded. For example, Data Store appears twice. The starting interface is indicated at the bottom rather than top of the workflow.

(11) Replacement of the Tuxedo package is mentioned briefly in the legend and in the figure but is presented without context. If important to the reader to know this, it should be described in text and put into context with best practices for RNA-Seq.

(12) Comparison with other tools: This section seems only to compare with Galaxy, are there other tools that should be included as the section title indicates?

(13) Since Galaxy is not the focus of the paper, figures should not be included depicting their interface. The snapshot of their workflow GUI is not very helpful in conveying the full picture.

(14) There are packages that exist to connect Galaxy and R now so this statement is not fully accurate.

(15) This section can be much more concise - describe the specific advantages of this system versus Galaxy, including access to genome resources, ability to store data, share data, etc. You have good points here but they are getting lost in delivery.

(16) Figures 3, 4, and 5 should be combined into 1 - possible with multiple parts (A, B, and C). They do not fit where currently referenced. It would make sense to reference these in the first section, after Figure 1. In addition, what is currently labeled as Figure 5 should be improved visually for clarity.

(17) Figure 6-10 are not appropriate. They are screenshots of tables of DE values that do not have context with a sample dataset and provide values that the reader cannot interpret. These would be best shown as a Venn diagram in a single figure (or a venn diagram or simple table of gene counts for both approaches). The tables of values from each R package should form supplemental tables.

(18) DiffExpress is mentioned for the first time farther down in the paper and in the figure legends. This should be clearly indicated in Figure 1 since it is important to the workflow described.

(19) An appropriate figure or table might involve a clear comparison of specific features and/or metrics between the two methods. It should also be made clear that this comparison is executed on the public instance of Galaxy - Galaxy can be installed locally to permit genome upload and abstraction for any command line tool or script.

(20) The way the sample dataset is used in the study does not make a great deal of sense since there is no known truth. Several tables of listed genes are shown only to say that there is a package that allows you to combine them together and find the intersection. Neither workflow can evaluate the study design and recommend an appropriate R package (or subset of R packages) - this would be something nice to have and worthy of a comparison. Otherwise, we are just listing numbers of genes, and more genes is not necessarily a better answer. I would re-structure this section and think about what you can compare and attempt to quantify this. I think it would relate more to features than to results from an experiment.

(21) The section labeled Discussion should be denoted as Conclusions

---

## Round 0.2 · Major Revisions

Dear Dr. Hubbard and colleagues:

Thanks for re-submitting your revision based on the reviewers’ comments. I received another round of review from one original reviewer, and they would like to see more done with your work.

While the concerns of the reviewer are relatively minor, this is a major revision to ensure that the original reviewer has a chance to evaluate your responses to their concerns.

I look forward to seeing your revision, and thanks again for submitting your work to PeerJ.

Good luck with your revision,

-joe

·

Basic reporting

(1) Proper literature references in introduction and for tools mentioned in the paper (HISAT2, Samtools), are not included in the current manuscript

Experimental design

no comment

Validity of the findings

no comment

Additional comments

Thank you for providing a revised version of the manuscript is improved however there are organizational details that remain a challenge. A few notes on the current text:

(1) The Introduction - more citations are needed in the Introduction - external support must be provided for most definitions as well as statements like "Although sequencing costs have decreased substantially for RNA-Seq, the computational footprint of RNA-Seq remains substantial." Many of these early statements could be quantified and would be more interesting to read. How many RNA-Seq studies have been completed? What does the increase look like over the past five years? These can be compiled from public data.

(2) p.2, line 97: "fRNAkenseq is among the first tool of its kind to employ API-oriented " Is it the first tool of its kind or? Define this statement a bit more.

(3) FASTQ should always be capitalized. Please review that all tools, packages, and file extensions are properly referenced and capitalized. They are inconsistently referenced through the text.

(4) Order of the Introduction will still cause issue for readers - make sure to define all terms before referencing your application, Cyverse, CoGE, Agave API, etc must be defined before talking about the tool discussed in this paper. That portion should come last.

(5) Workflow section: HISAT2 is listed as the mapper - does this take single and paired data - why only default parameter - what about species with very large introns? No customization possible?

(6) Proper manuscript citations are missing for tools such as HISAT2, Stringtie, Samtools, etc. Most of these public tools have a paper that goes with them.

(7) StringTie can detect novel isoforms, how is this handled downstream? It works by default with Ballgown - why is this not used downstream of StringTie?

(8) While I can appreciate that the authors will provide output from multiple R packages, there is literature that provides guidance on when specific packages should be used or should be used differently with different numbers of replicates. Since the replicate number and study design is known, some guidance should be provided by the application. This application is geared toward novice users with minimal training in HPC and R, therefore it is important to provide some suggested parameterization or at least suggestion of what subset of packages will be most appropriate given the design.

(9) A little more context is needed in the workflow section describing these tools - why did you choose these tools and parameters - provide some context and supporting literature here. Later, in the Demonstration, the venn diagram should be referenced to relate back to the importance of including these three tools. There is some discussion on this further in the paper but the workflow section should contain the general details on the full workflow, from start to finish. The support is still weak for why these three tools and I do believe it should only run the runs that are

(10) The workflow section does not mention CoGE but I assume this is how one obtains the version of the reference genome - are these pre-indexed for HISAT2?

(11) The Demonstration section starts with describing a chicken RNA-Seq dataset but the text reverts back to general usage (mentioning the ability to select one of 40,000 genomes etc in CoGE). This description should remain in the general workflow section. The Demonstration section needs to describe the data, the version of the genome used, the time for execution, etc. This section should demonstrate to us that this data is telling us something about how this program works and how well it works that could not be conveyed through a general description of the workflow.

(12) The portion on the synteny package for CoGE does not fit in this section. This could be included in the introduction, where it might be relevant to set the stage for CoGE initially. Synteny does not relate to RNA-Seq and is not appropriate later in the paper. The remaining part of this section should also likely go into workflow (where the genome browser visualization comes into view).

(13) The DiffExpress section should also be in the workflow section since this is a general description and not specific to the Demonstration set. For the Demonstration set, it would be relevant to discuss the study design and how that was passed to the tools (for example). Essentially, it seems like the Demonstration paragraph is sitting in the middle of the workflow section and breaking up that flow.

(14) The conclusion reads as though it is starting a new comparison with Galaxy. I would leave this to one to two sentences and instead remind the user of the overall utility of this tool. Focus in this section on reminding the user of the key features and advantages. Also, make sure we are aware how to start to using this and where to access it.

(15) The paper currently contains multiple misspellings, punctuation errors, and minor formatting issues. All co-authors should carefully review the document and correct these.

---

## Round 0.3 · accepted · Accept

Dear Dr. Hubbard and colleagues:

Thanks for re-submitting your revised manuscript to PeerJ, and for addressing the concerns raised by the reviewer. I now believe that your manuscript is suitable for publication. Congratulations! I look forward to seeing this work in print, and I anticipate it being an important resource for research communities studying large-scale transcriptomics datasets.

Thanks again for choosing PeerJ to publish such important work.

-joe